# Diffusion Approximation of Distribution Dynamics in an Agent-Based Economic Model with a Banking Sector

**Alexander S. Martynenko**
Sirius University
martynenko.as@talantiuspeh.ru

**Olga I. Krivorotko**
Sirius University
krivorotko.oi@talantiuspeh.ru

## Abstract

We study a discrete-time agent-based macroeconomic model with heterogeneous households, heterogeneous firms, and a stock–flow-consistent banking sector with explicit balance-sheet variables. The model includes interest accrual on deposits and loans, labor-market adjustment, production and wage payments, household consumption with credit access subject to borrowing constraints, firm credit financing, endogenous household and firm defaults, and bank balance-sheet updates with insolvency resolution.

The simulator produces aggregate time series for employment, output, consumption, prices, debt, deposits, defaults, and banking-regime indicators, together with cross-sectional summaries of household and firm states. These outputs support a distribution-dynamics perspective in which macroeconomic indicators are viewed as functionals of evolving agent-state distributions and the bank balance sheet.

Because interactions are mediated mainly through aggregates and credit constraints, the model can be interpreted as a controlled data-generating environment for studying when distribution-based or diffusion-style approximations capture macroeconomic dynamics, and when crisis regimes driven by defaults and banking distress require explicitly discrete treatment.

## 1 Introduction

Agent-based macroeconomic models (ABMs) have become an established complement to representative-agent and DSGE frameworks, particularly in settings where heterogeneity, credit constraints, default mechanisms, and balance-sheet interactions shape aggregate dynamics. Survey studies have organized the field's main directions and documented the ability of ABMs to reproduce business cycles, financial instability, and structural change emerging from micro-level interactions rather than from exogenous shocks (Dawid & Delli Gatti, 2018). Additional bibliometric and methodological reviews point to continued growth in macroeconomic ABM applications across areas such as macro-financial linkages, fiscal and monetary policy, and structural dynamics (Fagiolo & Roventini, 2017).

A significant strand of research links agent-based modelling with stock–flow-consistent (SFC) macroeconomics. In this tradition, detailed financial balance sheets are embedded within interacting-agent structures to ensure full accounting consistency across sectors, following the foundations laid out in Godley & Lavoie (2007). A widely used example is the ABM–SFC framework of Caiani et al. (2016), which introduces heterogeneous households and firms interacting with a banking sector subject to credit constraints and capital requirements, generating endogenous fluctuations and financial stress. Subsequent surveys have systematized developments in SFC macro modelling (Nikiforos & Zezza, 2017), while further applications extend these models to climate–finance and macro-financial feedbacks (Lamperti et al., 2018; Dafermos et al., 2017).

Parallel developments in mean-field and distribution-dynamics approaches provide tools for approximating high-dimensional stochastic agent systems through the evolution of state distributions. Mean field game (MFG) theory and related McKean–Vlasov formulations offer continuous-time, continuous-state representations of strategic interactions in large populations, with comprehensive

expositions provided by Carmona & Delarue (2018) and applications to finance and economic dynamics discussed in Carmona (2021). These frameworks suggest that, under appropriate conditions, large-scale agent-based dynamics may exhibit effective distribution-level behavior governed by drift–diffusion operators.

The model examined here is a discrete-time ABM featuring heterogeneous households, heterogeneous firms, and a single stock–flow-consistent banking sector. Its core mechanisms include: (i) interest accrual on deposits and loans; (ii) firms' labor-demand adjustment; (iii) production with wage payments and credit financing; (iv) household consumption subject to income-based borrowing limits and default thresholds; (v) firm defaults linked to leverage relative to a revenue proxy; and (vi) consistent balance-sheet updates for the banking sector. The model generates micro-to-macro time series for employment, output, consumption, prices, financial positions, default rates, and bank equity.

A further motivation is to use this ABM as a controlled environment for examining when distribution-based approximations—such as mean-field or diffusion-type limits—provide accurate representations of the underlying heterogeneous-agent dynamics. The evolution of cross-sectional balance-sheet distributions offers a natural basis for estimating effective drift and volatility components, while also enabling the analysis of settings where nonlinearities, threshold effects, and rare default cascades limit the applicability of continuous distribution-level approaches.

## 2   STRUCTURE OF THE ARTICLE

The article is organized as follows:

- Section 3 defines the implemented ABM, including agent sectors, parameter constraints, and the recorded state and outputs.
- Section 4 describes the numerical workflow: inputs, within-step update ordering, and exported time-series data.
- Section 5 reports representative simulation outputs, scenario-based experiments, and comparisons to differential-equation benchmarks.

## 3   PROBLEM FORMULATION: IMPLEMENTED AGENT-BASED ECONOMY

This section defines the implemented agent-based macroeconomic model, the admissible parameter domain, and the structure of simulated data used for analysis. The model operates in discrete time $t = 0, 1, 2, \ldots$ and consists of three sectors: heterogeneous households, heterogeneous firms, and a single stock–flow-consistent bank. All interactions occur through labor, product, and credit markets, and all monetary flows are recorded in balance-sheet variables.

### 3.1   ECONOMIC ENVIRONMENT

The economy contains:

- $N_H$ **households** indexed by $i$, each characterized by deposits $s_i(t)$, debt $d_i(t)$, employment status, realized income $y_i(t)$, and backward-looking expected income $\hat{y}_i(t)$.
- $N_F$ **firms** indexed by $j$, each with productivity $A_j$, cash holdings $m_j(t)$, debt $d_j(t)$, inventories $q_j(t)$, prices $p_j(t)$, markup $\mu_j(t)$, and labor demand $L_j(t)$.
- One **bank**, holding household and firm loans, deposits, reserves, and equity $K^B(t)$.

All monetary quantities in the model, such as deposits, debt, cash, inventories, and bank balance-sheet items evolve as explicit numerical state variables. Prices are endogenously set by firms through a unit-cost–markup rule; by contrast, wages and interest rates are fixed exogenous parameters in the current implementation (though they could be endogenized in future extensions).

### 3.2   PARAMETER DOMAIN AND CONSTRAINTS

Model parameters fall into several blocks.

**Household parameters.**

- Consumption propensity $\alpha_i \in (0,1)$ drawn from a truncated distribution (implementation uses bounded draws to avoid corner cases).
- Debt repayment fraction $\rho \in (0,1)$.
- Household debt-cap multiplier $\lambda_H > 0$, defining an income-based borrowing limit.
- Initial deposits and savings ranges specified by $[s_{\min}, s_{\max}]$.

Household borrowing at time $t$ is bounded by:

$$\ell_i(t) \leq \max\{0,\, \lambda_H \max(y_i(t), \varepsilon) - d_i(t)\},$$

and defaults occur when outstanding debt exceeds the income-based threshold. Negative deposits are not permitted, and consumption is truncated by available funds plus credit access.

**Firm parameters.**

- Productivity levels $A_j > 0$ drawn from a synthetic distribution.
- Wage level $w > 0$ (constant across firms).
- Adaptation rate $\gamma \in (0,1]$ for employment adjustment.
- Inventory sensitivity parameters for markup updates.
- Firm debt-cap multiplier $\lambda_F > 0$.
- Initial cash range $[m_{\min}, m_{\max}]$.

Firms may borrow to finance the wage bill if $m_j(t) < w\, L_j(t)$. In the current implementation, firm credit demand is rationed by (i) the bank-wide capital-based lending cap and (ii) an underwriting constraint expressed via a debt-service-ratio (DSR) limit. Separately, the parameter $\lambda_F$ enters the *default trigger*: a firm defaults when its debt exceeds a revenue-proxy threshold proportional to recent sales/revenue signals:

$$d_j(t) > \lambda_F\, \tilde{R}_j(t) \cdot \chi_F,$$

where $\tilde{R}_j(t)$ is a revenue proxy constructed from recent sales information (e.g., using $\max\{$last-period revenue, smoothed demand $\times$ price$\}$), and $\chi_F \geq 1$ is a default-trigger multiplier. No additional cash-flow condition is imposed in the current implementation.

**Bank parameters.**

- Loan interest rate $r_L \geq 0$ and a deposit-rate cap $r_D \geq 0$. The effective deposit rate $\tilde{r}_D(t)$ is set to zero when $K^B(t) \leq 0$, and otherwise scaled down so that total deposit interest paid does not exceed the realized loan-interest service actually collected in the same period.
- Credit multiplier $\kappa > 0$, establishing a capital-based lending cap:

$$L^{\text{used}}(t) \leq \kappa \max\{K^B(t), 0\},$$

  where $L^{\text{used}}(t)$ is the outstanding stock of loans (household + firm).
- Loss-given-default parameters $\text{LGD}_H, \text{LGD}_F \in [0,1]$ for household and firm defaults.
- Resolution parameters: recapitalization target ratio $r_{\text{recap}} \in [0,1)$ (deposit bail-in) and a switch for an optional external bailout to a minimum equity buffer $\underline{K}^B \geq 0$.

The bank absorbs realized credit losses directly through equity:

$$K^B(t+1) = K^B(t) + \Pi^B(t) - \text{Losses}(t),$$

where $\Pi^B(t)$ is the net interest margin.

**Insolvency resolution (implemented).** After balance-sheet update, if $K^B(t) < 0$ the model applies a resolution procedure. First, a *bail-in* is implemented via a proportional haircut on deposits (households and firms) to restore solvency and target a small positive equity buffer. Let total deposits be

$$D(t) = D_H(t) + D_F(t),$$

and let $X(t) \in [0, D(t)]$ denote the amount of deposits converted into bank equity. Deposits are reduced by $X(t)$ and equity is increased by $X(t)$:

$$K^B(t) \leftarrow K^B(t) + X(t), \qquad D(t) \leftarrow D(t) - X(t).$$

The conversion $X(t)$ is chosen so that the post-resolution equity satisfies a recapitalization rule

$$K^B(t) = r_{\text{recap}} D(t),$$

which yields

$$X(t) = \frac{r_{\text{recap}} D(t) - K^B(t)}{1 + r_{\text{recap}}} \quad \text{clipped to } [0, D(t)].$$

Equivalently, the implied haircut rate is $h(t) = X(t)/D(t)$ and deposits are multiplied by $(1-h(t))$.

If deposits are insufficient ($D(t) = 0$), or if equity remains negative after bail-in, the implementation can apply an *external bailout* (when enabled) that injects

$$B(t) = \max\{0, -K^B(t) + \underline{K}^B\}$$

to bring equity to the minimum buffer $\underline{K}^B$. If the bailout is disabled and equity remains negative, the bank enters a failure regime and new credit is not issued (available credit is set to zero).

### 3.3 STATE VECTOR AND DATA STRUCTURE

At each step, the simulator maintains a full cross-sectional microstate, including household deposits and debt together with income-history variables, firm cash, debt, inventories, prices and employment, and the bank balance-sheet stocks (loans, deposits, equity, reserves). From this evolving state it collects a consistent set of aggregate time series (real-side, price/markup, and financial indicators), together with event counts (defaults and bank-regime flags) and cross-sectional summaries such as quantiles. These recorded time series form the empirical foundation for downstream analysis, including distributional diagnostics and comparison with continuous-time macroeconomic approximations.

## 4 NUMERICAL METHOD AND SIMULATION WORKFLOW

This section outlines the numerical structure of the implemented agent-based model, including model inputs, the internal step-by-step update algorithm, and the structure of generated output data. The simulator operates in discrete time, and each step updates household, firm, and bank states according to deterministic rules possibly constrained by available liquidity, borrowing limits, and balance-sheet consistency.

### 4.1 MODEL INPUTS

A simulation run is fully determined by:

- **Population sizes:** number of households $N_H$ and firms $N_F$.
- **Household parameters:** consumption propensities, initial deposits, skill distribution, repayment fraction, and the income-based debt-cap multiplier (initial household debt is set to zero in the current implementation).
- **Firm parameters:** productivity distribution, initial cash levels, wage level, adaptation rate for labor adjustment, inventory parameters, and the firm debt-cap multiplier (initial firm debt is set to zero in the current implementation).
- **Bank parameters:** loan and deposit interest rates, the credit multiplier, and loss-given-default values for household and firm defaults.

- **Scenario configuration:** simulation horizon, random seed, and stress-mode settings (e.g., tighter credit constraints, higher leverage, or elevated LGD).

Synthetic initialization draws heterogeneous household and firm attributes from predefined distributions, ensuring replicability across runs.

## 4.2 Simulation Step: Internal Algorithm

Each period $t \to t+1$ follows a fixed sequence of operations. The ordering is essential, because decisions in later blocks depend on updated information from earlier ones.

**1. Interest accrual.** Outstanding household and firm debt accrues interest at rate $r_L$, and interest is *actually serviced* out of current deposits/cash up to the interest due; the bank's realized loan-interest cashflow in period $t$ is denoted by $I_L(t)$. Deposits accrue interest at an *effective* rate $\tilde{r}_D(t)$ that is bounded by the parameter $r_D$ but may be reduced in implementation: if $K^B(t) \leq 0$ then $\tilde{r}_D(t) = 0$, otherwise

$$\tilde{r}_D(t) = r_D \cdot \min\left\{1, \ \frac{I_L(t)}{r_D \, D(t)}\right\},$$

so that total deposit interest paid $I_D(t) = \tilde{r}_D(t) \, D(t)$ satisfies $I_D(t) \leq I_L(t)$. Net interest income updates bank equity by $K^B(t+1) = K^B(t) + I_L(t) - I_D(t)$ (before default losses and resolution).

**2. Labor-market adjustment.** Firms compute target employment $L_j^*(t)$ from expected demand and adjust their workforce partially toward this target (with bounded per-step adjustment). Hiring is limited by the pool of unemployed households; separations occur through the same partial-adjustment logic.

**3. Pricing, production, and wage payments.** Firms update prices via a unit-cost–markup rule with inventory-based adjustments. They compute wage obligations $wL_j(t)$ and request credit if cash-on-hand is insufficient. Production output is proportional to effective labor input. Firms also pay an overhead cost proportional to the wage bill; this amount is redistributed uniformly to households as transfers.

**4. Household consumption and demand allocation.** Each household forms desired consumption from deposits plus expected income and requests credit if desired consumption exceeds available funds. Actual consumption is limited by the available liquidity and borrowing bounds. Aggregate demand is distributed across firms using a discrete-choice allocation rule based on relative prices and (optionally) perceived quality. Optionally, unemployed households receive a fixed per-period transfer.

**5. Default events.** Households default if their debt exceeds an income-based threshold; Firms default when their debt exceeds a threshold linked to a revenue proxy; no additional cash-flow condition is imposed. Defaulted entities are reset according to the implemented logic, and default losses reduce bank equity. Firm default triggers a temporary exit (downtime) for a fixed re-entry lag; upon re-entry the firm restarts with inventories cleared and cash restored to at least a fraction of its initial cash endowment, and all workers are separated.

**6. Bank balance-sheet update.** The bank enforces accounting identities across loans, deposits, reserves, and equity. Credit supply is restricted by the capital-based multiplier; when equity becomes negative, the bank enters a failure regime in which no new lending is granted.

## 4.3 Numerical Characteristics

The model is deterministic conditional on the random seed. All updates occur synchronously within each block. No numerical integration or ODE/PDE solvers are used; all dynamics arise from iterative application of discrete rules over a large-dimensional state vector. The runtime scales approximately linearly in the number of households and firms.

## 4.4 SIMULATION OUTPUTS

At each step, the system records:

- household- and firm-level microstates (deposits, debt, cash, inventories, employment status);
- aggregate real-side variables (employment, unemployment rate, production, sales, consumption);
- price and markup statistics (firm-level prices, average price index);
- financial aggregates (household debt, firm debt, deposits, bank loans, bank equity) and resolution indicators;
- event metrics (household defaults, firm defaults, bank-failure indicator).

These time series constitute a complete data-generating process suitable for micro-to-macro analysis, stress testing, and comparison with continuous-time macro models.

## 4.5 EXPERIMENTAL WORKFLOW

Experiments consist of running the model under multiple scenarios and seeds:

- **baseline** configurations illustrating typical macro dynamics,
- **stress-credit** scenarios with tightened borrowing limits or increased LGD,
- **population-scaling** experiments for convergence analysis,
- **trend-matching** exercises comparing smoothed ABM trajectories with macro ODE benchmarks.

Outputs are exported as time-series files and used in downstream analysis of distribution dynamics and potential diffusion-style approximations.

## 5 NUMERICAL EXPERIMENTS AND SYNTHETIC DATA

This section documents the numerical experiments conducted with the implemented agent-based model, the structure of synthetic data produced by simulation, and comparisons with differential-equation macro models commonly used as aggregate benchmarks. We additionally relate our results to findings reported in the ABM and SFC macroeconomic literature.

## 5.1 SYNTHETIC DATA GENERATION

All experiments begin with the generation of heterogeneous household and firm populations. Initial conditions are drawn from user-specified distributions:

- household deposits from bounded uniform or lognormal distributions;
- firm productivity and initial cash from calibrated synthetic distributions.

The random seed ensures reproducibility across simulation runs. After initialization, the model produces step-by-step micro and macro time series recording employment, consumption, production, prices, debt, deposits, default counts, and banking-sector variables.

## 5.2 REPRESENTATIVE FIGURES

To demonstrate typical qualitative regimes, we report a compact figure set generated from a representative run of the current implementation. The showcased configuration is chosen to exhibit non-trivial credit dynamics and occasional default or bank-regime events while remaining numerically stable. In these plots, `Employment` denotes the number of employed households, `Output` denotes realized sales in physical units (inventory-constrained), and `Consumption` denotes aggregate consumption expenditure. The bank-regime figure reports bank equity together with resolution/bailout/failure indicators recorded by the simulator.

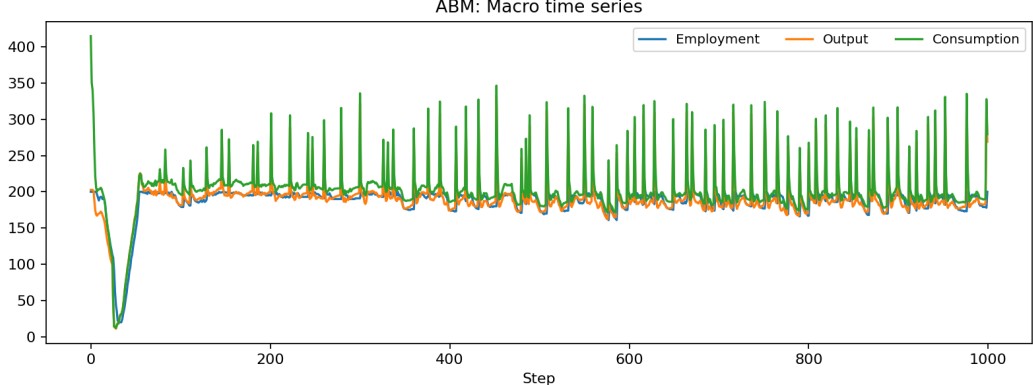

Figure 1: Representative macro time series: employment, output (sales in units), and consumption expenditure.

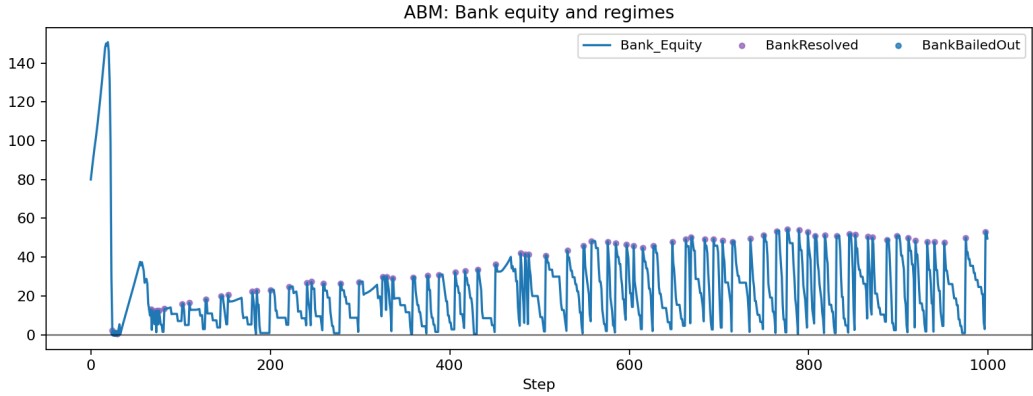

Figure 2: Bank equity and regime events: resolution (bail-in), bailout (if enabled), and failure indicator.

## 5.3 BASELINE EXPERIMENTS

A baseline configuration is used to illustrate typical macroeconomic dynamics. The model generates:

- convergence of aggregate employment toward a fluctuating steady region,
- co-movement of consumption and output driven by household borrowing constraints,
- price adjustments resulting from inventory pressure and variations in firm-level cash flow,
- persistent heterogeneity in household and firm balance-sheet positions,
- irregular but recurrent default events feeding back into bank equity.

Figure 1 provides a representative view of the resulting macro time series. The resulting trajectories resemble stylized business-cycle patterns reported in earlier ABM–SFC models such as Caiani et al. (2016) and in more recent macro-financial ABM validations summarised by Dawid & Delli Gatti (2018).

## 5.4 STRESS-CREDIT SCENARIOS

Stress experiments tighten borrowing limits (lower household or firm debt caps), increase loss-given-default, or reduce bank capital. These scenarios generate:

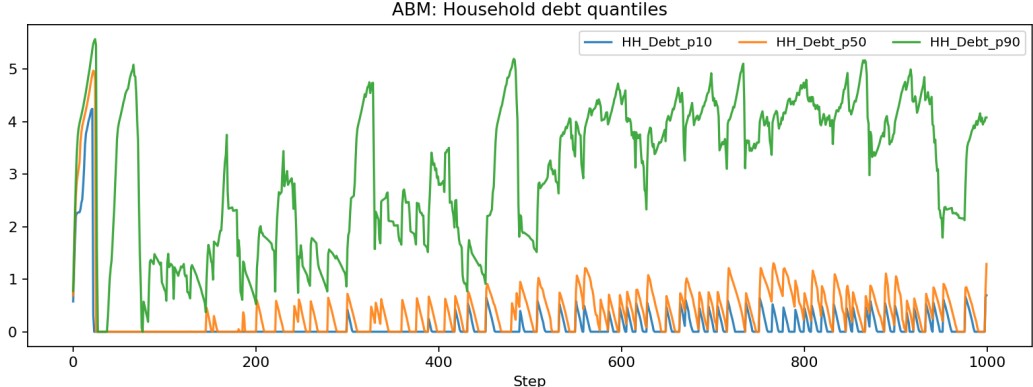

Figure 3: Distributional diagnostics: household debt quantiles over time.

- stronger default clustering among constrained households and firms,
- sharper declines in bank equity,
- temporary collapses in credit supply,
- amplified output contractions followed by partial recoveries.

Bank equity dynamics and resolution indicators are illustrated in Figure 2, while distributional tail behavior is summarized by the quantile plot in Figure 3. Such dynamics closely mirror stress behaviors discussed in ABM-based macro-financial studies (e.g., Lamperti et al. (2018); Dafermos et al. (2017)), where credit tightening and balance-sheet deterioration induce nonlinear adjustments in aggregate activity.

### 5.5    POPULATION-SCALING AND CONVERGENCE CHECKS

To examine robustness of macro outcomes, we run simulations across increasing population sizes $(N_H, N_F)$ and multiple random seeds. We record:

- convergence of smoothed aggregate variables (output, employment, average price),
- persistence of distributional heterogeneity (quantiles of household debt, firm leverage),
- reduction in Monte Carlo variance of macro time series as system size grows.

The results align with findings from the ABM convergence literature that aggregate trajectories tend to stabilize under large populations, while distributional tails remain informative for stress dynamics and rare default events.

### 5.6    COMPARISON WITH DIFFERENTIAL-EQUATION BENCHMARKS

We compare smoothed ABM trajectories with continuous-time macroeconomic benchmarks such as the Solow growth model. Following a purely descriptive approach, we:

1. construct an aggregate output-per-worker series from ABM data as $Y(t)/L(t)$,
2. apply a moving-average smoothing window,
3. fit Solow-type ODE trajectories $k(t)$ to match the shape of the smoothed ABM trend.

This comparison shows that ABM data admit a low-frequency trend compatible with standard macro growth equations, while short-run dynamics display volatility, discontinuities, and regime shifts (e.g., bank-capital failures) that cannot be represented within deterministic ODE frameworks. Similar interpretations are discussed in Lamperti et al. (2018) and in recent macro-ABM forecasting work Di Domenico et al. (2025). The descriptive trend fit and its absolute-error diagnostic are shown in Figures 4 and 5.

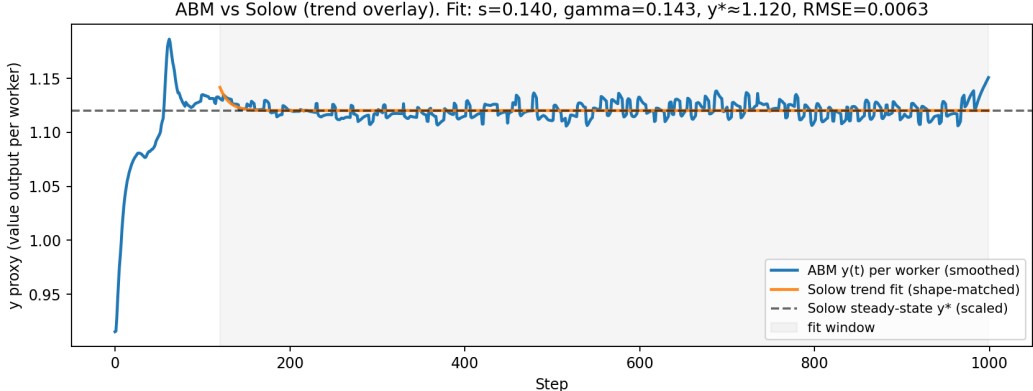

Figure 4: Low-frequency comparison: a smoothed ABM output-per-worker proxy against a fitted Solow-type trend (descriptive fit).

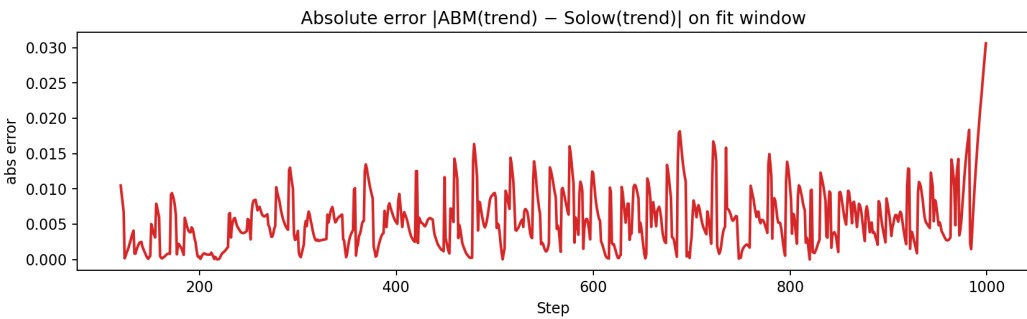

Figure 5: Diagnostic for the descriptive fit: absolute error between the smoothed ABM proxy and the fitted Solow-type trend.

## 5.7    SUMMARY OF OBSERVED DYNAMICS

Across all experiments, the model consistently produces:

- realistic micro-to-macro transmission mechanisms driven by credit constraints,
- endogenous default clustering and bank-capital fluctuations,
- persistent dispersion in household and firm balance-sheet distributions,
- smooth aggregate trends overlaying non-smooth micro-induced volatility.

These features position the model as a suitable data-generating process for future exploration of distribution-dynamics and diffusion approximations.

## 6    CONCLUSION

This paper has presented a consolidated description of a discrete-time agent-based macroeconomic model integrating heterogeneous households, heterogeneous firms, and a stock–flow-consistent banking sector. The implemented model combines income-based borrowing limits, balance-sheet constraints, endogenous pricing with inventory signals, labor-demand adaptation, and explicit handling of household and firm defaults. Numerical experiments show that the model consistently generates: (i) realistic micro-to-macro transmission from household and firm balance-sheet conditions to aggregate output and employment, (ii) endogenous clustering of defaults linked to credit tightening and bank-capital fluctuations, and (iii) heterogeneous and persistent cross-sectional distributions of financial variables that shape aggregate dynamics beyond mean averages.

A broader motivation of this work is to position the ABM as a controlled environment for studying when continuous distribution-level approximations may capture the essential dynamics of a heterogeneous economy. Because the model produces full cross-sectional distributions of household and firm states, it offers a natural dataset for evaluating potential mean-field or diffusion-type descriptions of macroeconomic evolution. While we do not derive a formal diffusion limit or McKean–Vlasov equation in the present work, the observed separation between smooth aggregate trends and volatile micro-driven fluctuations suggests that effective drift–diffusion components may emerge under appropriate scaling or smoothing. Identifying such conditions is a key step toward bridging ABM simulations with mean-field and distribution-dynamics methodologies.

The present paper therefore provides both a self-contained documentation of a functioning ABM and a roadmap for developing distribution-based analytical tools grounded in fully specified micro-level dynamics.

## 7 Future Work and Extensions

Future work will proceed along three directions. First, we will formalize an empirical distribution-dynamics framework by extracting drift and volatility terms from simulated cross-sectional data, enabling a diffusion-style surrogate model. Second, we will study the validity limits of such approximations, focusing on nonlinearities introduced by credit constraints, inventory-based pricing, and discrete default thresholds. Third, we will extend the model toward richer institutional structures—explicit bank-resolution regimes, multi-bank competition, or adaptive wage-setting—while preserving the accounting consistency that characterizes the current implementation.

## Acknowledgements

This research was supported by the grant of the Federal Territory "Sirius" under the state program *Scientific and Technological Development of the Federal Territory "Sirius"* (Agreement No. 26-03, dated 07 July 2025).

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
