# OpenReview forum: "Diffusion Approximation of Distribution Dynamics in an Agent-Based Economic Model with a Banking Sector"
_mathai.club/MathAI/2026/Conference — 2026 Oral_

### Official Review · Reviewer_H1Hd · 2026-03-10
**This paper contributes to the development of agent-based modeling as a controlled environment for the study of a heterogeneous economy, combining the micro and macro levels.**

**Rating:** 8
**Confidence:** 3

**Review:**

Quality
This paper documents a sophisticated agent-based macroeconomic model that integrates heterogeneous households, firms, and a stock-flow-consistent banking sector. The work is technically rigorous, clearly presented, and well-grounded in the relevant literature (ABM, SFC, and mean-field game theory).
Clarity
The text of the paper presents the formulation of the model, describes all the main components, parameters and mechanisms. Thus, the presentation of the model is clear and transparent.
Originality
The paper's originality lies in its explicit positioning as a «controlled environment» for studying the limits of distribution-based approximations (like mean-field or diffusion limits) for macroeconomic dynamics.
Significance
This paper contributes to the development of agent-based modeling as a controlled environment for the study of a heterogeneous economy, combining the micro and macro levels.
Core Strengths:
•	Built on a stock-flow consistent framework with transparent rules for borrowing, defaults, and bank resolution.
•	Comprehensive setup includes baseline runs, stress tests and experiments that demonstrate the model's range of behaviors.
•	Reproducibility due to the detailed description of the model, parameters and steps.
Main Limitations:
•	Key variables (wages, interest rates), as well as rules are fixed and exogenously set.
•	Lack of calibration based on real data or rigorous quantification of convergence or crisis dynamics.
The article offers a well-designed description of a model that can potentially contribute to the development of agent-based modeling in macroeconomics. In future studies, this model can be complicated and expanded, as well as brought closer to real data through calibration.

---

### Official Review · Reviewer_HaT4 · 2026-03-13
**Clear ABM documentation, but limited novelty, incomplete delivery on the diffusion promise, and a double-blind formatting issue**

**Rating:** 3
**Confidence:** 3

**Review:**

### Quality, clarity, originality
This paper presents a discrete-time agent-based macroeconomic model with heterogeneous households, heterogeneous firms, and a stock-flow-consistent banking sector. The simulator description is competent and the manuscript is structured enough to follow. However, as a research paper, the work currently feels more like a model note / implementation report than a completed scientific contribution.

My main concern is that the delivered contribution is substantially narrower than the title and framing suggest. The manuscript does not  derive, estimate, or validate a diffusion approximation, a McKean–Vlasov limit. The conclusion explicitly states that the paper does not derive a formal diffusion limit in the present work. As a result, the work reads primarily as a structured description of an implementation plus illustrative simulations, rather than as a paper that materially advances diffusion-based approximation of ABM dynamics.

### Significance
Potentially moderate, currently limited.

### Strengths
- The paper is relatively clear at the level of model mechanics. The household, firm, and bank sectors are described in enough detail.
- The manuscript gives a coherent step-by-step simulation workflow.
- The experiments cover several scenario types (baseline, stress-credit, scaling, descriptive trend matching).
### Weaknesses
- Novelty is weak in the current version. The paper does not yet provide the core scientific contribution implied by the title, namely a genuine diffusion-oriented approximation or analysis of distribution dynamics beyond qualitative motivation.
- The experiments are mostly illustrative.
- The manuscript contains an unresolved placeholder citation at the very bottom of page 9.
- The title on openreview differs from the one in the pdf file itself.
- The paper is marked as “Anonymous authors” and “under double-blind review,” but it still contains an acknowledgements section with a named funding source and a specific agreement number and date.

---

### Official Review · Reviewer_7Bxe · 2026-03-13
**Well-documented model with faint delivery**

**Rating:** 5
**Confidence:** 3

**Review:**

The paper presents a detailed description of a discrete-time agent-based macroeconomic model with heterogeneous households, heterogeneous firms, and a stock–flow consistent banking sector. The model includes balance-sheet variables, credit constraints, default mechanisms, and a banking resolution procedure. The authors provide a clear algorithmic description of the simulation workflow, including the ordering of update steps, parameter domains, and the structure of generated micro- and macro-level time series.

The work is well documented and provides a reproducible framework for generating synthetic macroeconomic data from a heterogeneous-agent system. Numerical experiments illustrate typical macroeconomic regimes, stress scenarios, and scaling behavior with respect to population size. The authors also compare smoothed aggregate outputs with simple differential-equation benchmarks and discuss possible connections to distribution-dynamics and diffusion-type approximations.

Overall, the paper offers a self-contained description of a functioning ABM implementation and may serve as a useful computational environment for studying distributional dynamics or testing approximation methods in macroeconomic modeling.

---

Strengths:
1. Topic relevance is discussed and the paper provides a clear historical and methodological context for agent-based macroeconomic modeling.
2. The model structure and simulation workflow are described in detail, supporting reproducibility of experiments.
1. The paper presents sufficient experimental assessment including baseline simulations, stress scenarios, and population scaling experiments. The framework may serve as a useful data-generating environment for evaluating future mean-field or diffusion approximations.

---

Other notes:
1. From the perspective of mathematical modeling, the contribution is relatively modest. The paper mainly documents an implementation of an agent-based model rather than introducing new analytical methods. The simulation step relies on deterministic update rules, with only initial state conditioned on the random seed, which limits the methodological novelty from a modeling perspective. Section 4.2 (simulation algorithm) should be more explicitly linked to the equations and mechanisms introduced earlier in the model description. Additional references would strengthen the justification of the chosen distributions and transition functions used in the simulation.
2. Although the title emphasizes diffusion-oriented approximation, the paper does not actually derive such approximations; it only discusses them as potential future work.
4. The abstract should be shortened for the camera-ready version.

The paper provides a clear description of an agent-based macroeconomic model and may be useful as a computational framework for studying distributional dynamics. However, the main contribution lies more in economic modeling and simulation infrastructure than in new mathematical or AI-oriented methods. The announced connection to diffusion or mean-field approaches is not developed in the current version and remains a direction for future work.
For this reason, while the work is technically sound, it appears only weakly aligned with the main themes of the conference on mathematical methods in AI.

---

### Official Review · Reviewer_jiJ9 · 2026-03-13
**A Well-Documented ABM Framework with Promising Motivation but Limited Diffusion Delivery**

**Rating:** 6
**Confidence:** 4

**Review:**

This paper presents a discrete-time agent-based macroeconomic model with heterogeneous households, heterogeneous firms, and a stock-flow-consistent banking sector. The model incorporates explicit balance-sheet variables, credit constraints, endogenous defaults, and a bank resolution mechanism, and it is positioned as a controlled environment for studying distribution dynamics and, potentially, diffusion-style approximations of heterogeneous-agent macroeconomic systems.

Overall, I found the paper interesting and reasonably well executed. Its main strength lies in the clarity and completeness of the model description rather than in a new mathematical result. The manuscript gives a structured account of the economic environment, parameter constraints, simulation workflow, and generated outputs, which makes the framework relatively transparent and potentially useful for future work on distribution-based approximations. In that sense, I believe the paper is slightly above the acceptance threshold, although the current version still falls short of fully delivering the contribution suggested by its title.

The main weakness is that the actual scientific contribution is narrower than the title and framing initially suggest. Despite the emphasis on “diffusion-oriented approximation” and distribution dynamics, the paper does not derive, estimate, or validate a formal diffusion approximation, McKean–Vlasov limit, or related reduced-form analytical model. Instead, it mainly motivates such a direction and presents the ABM as a possible foundation for future work. As a result, the manuscript currently reads more as a strong implementation and modeling note than as a completed contribution on diffusion-based approximation itself.

There are also several presentation issues that should be addressed in a revised version. The manuscript still contains signs of incompleteness, including an unresolved placeholder citation in the benchmarking discussion, and the double-blind formatting is not fully clean because the acknowledgements reveal a specific funding source.

Overall, my assessment is positive but cautious. I think the paper offers a well-documented and potentially useful ABM framework for studying macroeconomic distribution dynamics, and this gives it enough value to be considered marginally above the acceptance threshold. However, the current version does not yet fully deliver on the diffusion-oriented promise highlighted in the title, and its main contribution remains in modeling infrastructure and simulation design rather than in a genuinely developed approximation method.

---

### Decision · Program_Chairs · 2026-03-14

**Decision:**

Accept (Oral)

**Comment:**

Dear Author(s),

On behalf of the Program Committee of the International Conference on Mathematics of Artificial Intelligence (MathAI 2026), we are pleased to inform you that your paper has been accepted for an oral presentation at MathAI 2026.

Your paper was evaluated through a rigorous two-stage review process involving both automated screening and expert review by members of the Program Committee. The reviewers recognized the quality and contribution of your work.

Presentation details:

- Format: Oral presentation (15–20 minutes + 5 minutes Q&A)
- Mode: You may present either in person (offline) at the conference venue in Sirius, Russia, or remotely via Zoom. Please indicate your preferred mode when confirming your participation.
- Conference dates: Marh 30 - April 3, 2026
- Website: https://mathai.club

Next steps:

1. Please confirm your participation and presentation mode by replying to this email mathai.club@yandex.ru no later than March 15, 2026 18:00 Moscow time.
2. If you plan to attend in person, the organizing committee will provide accommodation details separately.
3. Please prepare your final camera-ready manuscript according to the formatting guidelines available at https://mathai.club and upload it to OpenReview by March 15, 2026 18:00 Moscow time.

Should you have any questions regarding the program, logistics, or your presentation slot, please do not hesitate to contact us.

We look forward to your contribution to MathAI 2026.

With kind regards,

MathAI 2026 Program Committee
International Conference on Mathematics of Artificial Intelligence
https://mathai.club
OpenReview: https://openreview.net/group?id=mathai.club/MathAI/2026/Conference
Telegram: https://t.me/MathAI_club
Email: mathai.club@yandex.ru